# Fabrication of an Extremely Cheap Poly(3,4-ethylenedioxythiophene) Modified Pencil Lead Electrode for Effective Hydroquinone Sensing

**DOI:** 10.3390/polym13030343

**Published:** 2021-01-22

**Authors:** Jian-Yu Lu, Yu-Sheng Yu, Tung-Bo Chen, Chiung-Fen Chang, Sigitas Tamulevičius, Donats Erts, Kevin C.-W. Wu, Yesong Gu

**Affiliations:** 1Department of Chemical and Materials Engineering, Tunghai University, No.1727, Sec.4, Taiwan Boulevard, Xitun District, Taichung 40704, Taiwan; s06310152@go.thu.edu.tw (J.-Y.L.); tchen4@thu.edu.tw (T.-B.C.); 2Department of Chemical Engineering, National Taiwan University, No. 1, Sec. 4, Roosevelt Road, Taipei 10617, Taiwan; b04504074@g.ntu.edu.tw (Y.-S.Y.); kevinwu@ntu.edu.tw (K.C.-W.W.); 3Department of Environmental Science and Engineering, Tunghai University, No.1727, Sec.4, Taiwan Boulevard, Xitun District, Taichung 40704, Taiwan; cfchang@thu.edu.tw; 4Institute of Materials Science, Kaunas University of Technology, 59 K. Barˇsausko St., LT-51423 Kaunas, Lithuania; sigitas.tamulevicius@ktu.lt; 5Institute of Chemical Physics, University of Latvia, 19 Raina Blvd., LV-1586 Riga, Latvia; donats.erts@lu.lv

**Keywords:** poly(3,4-ethylenedioxythiophene), pencil lead, graphite, hydroquinone, biosensor

## Abstract

Hydroquinone (HQ) is one of the major deleterious metabolites of benzene in the human body, which has been implicated to cause various human diseases. In order to fabricate a feasible sensor for the accurate detection of HQ, we attempted to electrochemically modify a piece of common 2B pencil lead (PL) with the conductive poly(3,4-ethylenedioxythiophene) or PEDOT film to construct a PEDOT/PL electrode. We then examined the performance of PEDOT/PL in the detection of hydroquinone with different voltammetry methods. Our results have demonstrated that PEDOT film was able to dramatically enhance the electrochemical response of pencil lead electrode to hydroquinone and exhibited a good linear correlation between anodic peak current and the concentration of hydroquinone by either cyclic voltammetry or linear sweep voltammetry. The influences of PEDOT film thickness, sample pH, voltammetry scan rate, and possible chemical interferences on the measurement of hydroquinone have been discussed. The PEDOT film was further characterized by SEM with EDS and FTIR spectrum, as well as for stability with multiple measurements. Our results have demonstrated that the PEDOT modified PL electrode could be an attractive option to easily fabricate an economical sensor and provide an accurate and stable approach to monitoring various chemicals and biomolecules.

## 1. Introduction

Hydroquinone (HQ) is considered one of the main benzene metabolites detected in human urine and blood, which could also be generated from chemicals of other sources, such as automobile exhaust, dietary intake, and cigarette smoke [1,2,3,4,5,6,7]. In humans, HQ is often found to form specific complexes with proteins that could interfere with the biological functions of proteins or induce the accumulation of oxidative stress that is inevitably responsible for DNA damage, apoptosis, cytotoxicity, and tumorigenesis [8,9,10,11,12,13]. Long-term exposure to HQ has been deliberated to play an important role in organ dysfunction, leukemia pathogenesis, and cancer formation [13,14]. Therefore, tracing HQ in human urine and blood is strongly recommended in the pollution-intensive industrial society, especially for benzene-exposed workers.

Hydroquinone (HQ) in human urine and blood can be accurately measured in the laboratory with expensive instruments, such as the gas chromatography-mass spectrometry method [15], synchronous fluorescence technique [16], coupled-column HPLC with fluorimetric detection [17], HPLC [18,19], flow-injection chemiluminescence (CL) method [20], capillary electrophoresis/electrospray ionization-mass spectrometer [21], and others. Notably, these methods are apparently inconvenient for routinely monitoring workers to avoid possible occupational benzene poisoning. Since HQ possesses anti-oxidative property, it can be easily oxidized to catechol and vice versa. Therefore, electrochemical methods have shown great promise to precisely detect both hydroquinone and catechol [14,22,23,24,25]. Although numerous electrodes have been fabricated to achieve the sensitive quantification of hydroquinone and catechol, there is still some room for improvement regarding feasible construction and cost reduction. In order to replace the relatively expensive metal or glassy carbon electrodes, the use of extremely cheap and disposable pencil lead (PL or graphite), commonly used in writing instruments, was attempted to construct more environmentally friendly electrodes.

Recently, a variety of modified pencil graphite electrodes have shown great promise for the detection of folic acid, ascorbic acid, hydrazine, nicotine, as well as DNA and RNA [26,27,28,29,30]. Since pencil lead is usually made from a mixture of graphite and clay, some unexpected electrochemical signals might inevitably appear while directly using the pencil lead as the working electrode, we, therefore, attempted to coat the pencil lead with the conductive polymer, poly(3,4-ethylenedioxythiophene) or PEDOT. Generally, PEDOT film is facilely fabricated on an electrode with an electrochemical procedure, and its well-defined surface morphology is suitable for the immobilization of biomolecules. In particular, the sulfur atoms in the PEDOT molecular structure open up the opportunity to stably capture the golden nanoparticles, which are able to immobilize a variety of biomolecules with thiol groups (-SH), such as proteins and nucleotides, to construct various biosensors [31,32,33]. Previously, we have also demonstrated that modification of a platinum electrode with the conductive PEDOT film could effectively eliminate the hydrogen ion reduction and water oxidation at slightly negative or positive potentials [33], which provided limited background interference for a sensitive detection [32,34,35,36].

In this study, we attempted to modify an extremely inexpensive pencil lead electrode with the electrochemically modified PEDOT, named PEDOT/PL electrode, in order to accomplish stable and sensitive sensing of hydroquinone. For this purpose, we characterized the PEDOT film on the PL electrode and its stability for multiple measurements. The linear correlation between the electrochemical signal and the concentration of HQ was established with different methods, such as cyclic voltammetry (CV) and linear sweep voltammetry (LSV). In addition, we explored the influences of pH, temperature, scan rate of voltammetry, film thickness of PEDOT, as well as some components in human blood, on the performance of PEDOT/PL in sensing HQ.

## 2. Material and Methods

### 2.1. Chemicals

Lithium perchlorate trihydrate (LiClO_4_) used as an electrolyte for PEDOT synthesis was the product of Merck KGaA at Darmstadt, Germany. Monomer EDOT was bought from Sigma-Aldrich Corp. at St. Louis, MO, USA, A piece of 2B pencil lead in 0.5 mm diameter was available from a convenient stationery office. All other chemical reagents from different sources were of analytical grade.

### 2.2. Construction of a PEDOT/PL Electrode

A CHI-621B electrochemical analyzer (CHI instruments at Austin, TX, USA) was routinely utilized for electrochemical analysis of PEDOT film and subsequent detections. To prepare the working electrode, a PL with a length of 1.5 cm and a diameter of 0.6 mm was first polished with white print paper to remove the surface wax, then one centimeter of the PL was submerged into a solution containing 0.1 M of LiClO_4_ and 0.01 M of EDOT. To electrochemically synthesize PEDOT film on the PL, a bare pencil lead and an Ag/AgCl (3 M NaCl) electrode served as auxiliary and reference electrodes, respectively, while the potential was swept from 0.0 to 1.0 V with a scan rate of 100 mV·s^−1^ for six cycles under an ambient environment, around 25 °C. The constructed PEDOT/PL electrode was soaked in a 0.1 M phosphate-buffered saline buffer (PBS, pH 7.4) for 10 minutes to eliminate possible unreacted EDOT and was gently rinsed with the PBS buffer three times, then was stored in a desiccator under room temperature for further experiments.

### 2.3. Characterizations of PEDOT/PL

The FE-SEM images and the EDS mapping of the PL and PEDOT/PL electrodes were taken by the FEI Nova NanoSEM 230 from Thermo Fisher Scientific (Hillsboro, OR, USA). The sample was taped on the carbon conductive tape then kept under vacuum for one day to remove the moisture. Without being sputtered by conductive elements, the sample was placed into the electron microscope for observation. During the operation, the accelerating voltage was set as 10 kV, and the EDS mapping was performed by collecting the Kα signals of carbon, oxygen, and sulfur over 150 s. The FTIR spectra of PL and PEDOT/PL electrodes were obtained on an IRTracer-100 Fourier Transform Infrared (FTIR) Spectrophotometer from Shimadzu (Kyoto, Japan). The sample was placed on the surface of an ATR crystal plate in the high-pressure single reflection MIRacle ATR system for scanning in the wavenumber range of 650–4000 cm^−1^, and the ATR spectrum was then processed by the ATR correction program for better band intensities.

### 2.4. Measurement of Hydroquinone by the Constructed PEDOT/PL Electrode

All electrochemical measurements were conducted in a miniature electrochemical cell with a modified PEDOT/PL electrode as the working electrode, a bare pencil lead as the auxiliary electrode, and an Ag/AgCl (3M NaCl) electrode as the reference electrode. The cyclic voltammetry (CV) and linear sweep voltammetry (LSV) were carried out in 6 mL of 100 mM PBS containing hydroquinone with pH about 7.4 under room temperature. The scan rate was 100 mV·s^−1^ and the potential window was from −0.5 to 0.7 V. During the 21-day period experiment, the electrode was gently washed with distilled water three times and stored at room temperature for the next test.

## 3. Experimental Results

### 3.1. Biosensor Fabrication and Characterizations

It has been reported that the PEDOT coated on the platinum electrode might be the best candidate for the electrochemical detection of hydroquinone due to its intrinsic capability for electron transfer [37]. However, we have demonstrated that the direct coating of electrochemically synthesized PEDOT film on the Pt electrode often leads to film crack during multiple measurements [31,33]. In this study, we successfully modified a pencil lead (PL) electrode with the electrochemically synthesized PEDOT film. As shown in Figure 1, we could clearly notice the interface of PEDOT film and the uncovered pencil lead. After enlarging the images, we were able to distinguish the surface morphology difference between pencil lead and PEDOT film such as pieces of graphite on the pencil lead and small granules on the PEDOT film (Appendix A). The EDS analysis has revealed that the pencil lead was predominately carbon element, whereas the oxygen and sulfur elements arose on the PEDOT film besides the carbon elements. The distribution of oxygen and sulfur elements that originated from PEDOT could also be observed by element mapping, as shown in Figure 1B–D, where the interface was clearly noticed, and oxygen and sulfur elements shined on the PEDOT film. The ratio of O/S for PEDOT film was approximately 3.1, which was greater than 2 according to the molecular structure of PEDOT, possibly due to the exposure of the electrode to the atmosphere. In addition, the excess carbon element for PEDOT film implicated the super thin layer of the PEDOT from six cycles of cyclic voltammetry synthesis, which might be related to the general detection depth, approximately 1–3 µm, in EDS analysis. In conclusion, the conductive PEDOT film was effectively synthesized on the PL electrode with the electrochemical approach.

### 3.2. Characterization of PEDOT/PL by FTIR

FTIR is commonly employed to track the surface modification of materials. Figure 2 displays the FTIR spectra of PL and PEDOT/PL, which also demonstrated the successful synthesis of PEDOT on the PL electrode. For the bare PL electrode (line a), we noticed the significant noise that might have resulted from the other ingredients in the pencil lead from a convenient stationery office, such as clay and wax. Nevertheless, we could still observe the broad vibrational bands in the range of 2500–3300 and around 1430 cm^−1^ in corresponding to the CH_2_ stretching [38,39]. Besides, some characteristic bands related to oxygen functional groups were found at 3500, 1580, 1270, and 1072 cm^−1^, which might also contribute to the higher O/S element ratio in EDS analysis. Lines b demonstrated the typical spectra of PEDOT, which including asymmetric stretching vibrations of C=C and C–C in the thiophene ring at 1528 and 1372 cm^−1^, bend vibration of C–O–C in the ethylenedioxy ring at 1220 and 1062 cm^−1^, stretching vibrations of C–S–C bond in the thiophene ring at 980, 850, and 696 cm^−1^, and possible deformation of the ethylenedioxy ring at 920 cm^−1^ [35,40,41]. The FTIR spectrum of PEDOT remained almost the same after HQ measurement as shown by line c, indicating the good stability of PEDOT film on the PL electrode.

### 3.3. Detection of Hydroquinone with a PEDOT/PL Electrode

Cyclic voltammetry (CV) is a sophisticated electrochemical technology used to provide convenient and precise measurement of target chemical compounds. Figure 3 displays that the CV profile of a bare PL electrode in response to HQ, where the anodic peak demonstrated the oxidation of hydroquinone to benzoquinone and the cathodic peak exhibited its reversed reduction of benzoquinone to hydroquinone. The reductive and oxidative mechanism was shown in Appendix A. Based on the results, we could learn that the PEDOT/PL electrode had the potential for the electrochemical measurement of HQ, but the bare commercial pencil lead could only provide a CV profile with relatively flat and more separated redox peaks. In comparison, the modification of PL electrode by PEDOT film could make the redox peaks closer and sharper with higher intensity, indicating the role of PEDOT film in reducing redox potentials, accelerating electron transfer, and minimizing the adsorption of HQ and its reduced product on the electrode.

Figure 4A shows the cyclic voltammetry profile of different concentrations of hydroquinone on the PEDOT/PL electrode, where well-defined anodic and cathodic peaks are located at the potential of near 0 V to 0.1 V, respectively. Although the PEDOT brought the anodic and cathodic peaks closer, the separation was still larger than the value of 28 mV for a normal two-electron reversible process and increased by increasing the concentration of hydroquinone. In addition, the average ratio of anodic peak current (*i*_pa_) and cathodic peak current (*i*_pc_) was slightly larger than unity, for example, *i*_pa_/*i*_pc_ = 1.36 for 6 mM, both indicating the inevitable influences from the diffusion resistance and possible adsorption of hydroquinone and benzoquinone on the electrode during the cyclic voltammetry measurement. Figure 4B presents a nice linear correlation between the anodic peak current and the concentration of hydroquinone, where the slope of linear correlations for concentration ranges of 2 to 10 mM was 0.1272 μA·μM^−1^. Considering the surface area of this PEDOT film was approximately 0.1719 cm^2^, the sensitivity of the PEDOT/PL electrode in sensing hydroquinone would be 734 μA·mM^−1^·cm^−2^. The inlet figure also revealed a good linear correlation for lower concentrations of HQ in the range of 10–100 μM with a slope of 0.3404 μA·μM^−1^ and sensitivity of about 1980.2 μA·mM^−1^·cm^−2^. Based on the standard deviation of the peak current, the limit of detection (LOD) of this study was approximately 7.7 μM, which was compatible with other approaches [42].

### 3.4. The effect of pH on the Detection

Considering the pK_a_ of HQ is about 9.85, it was reported that pH near or higher than the pK_a_ might lead it to easy deprotonation [34]. We, therefore, investigated the performance of the PEDOT/PL electrode in response to 6 mM of HQ in 0.1 M PBS buffer with different pH. In Figure 5A, we clearly observed that the measurement preferred pH 7.4 to other pH examined. While increasing the pH from 5.9 to 7.4, the HQ was deprotonated more easily, however, too much hydroxyl ions would decrease the adsorption of HQ on the PEDOT film and therefore the peak current [34]. According to Figure 5B, the oxidation potential of HQ decreased from 0.3 V to 0.1 V by increasing pH from 5.9 to 8.9, which was ascribed to the easiness of HQ deprotonation under a relatively higher pH. On the other hand, the oxidation of HQ under the pH below its pk_a_ implicated that the oxidation of HQ might be also catalyzed by coupling of the intrinsic reduction of PEDOT film. If we applied the PEDOT/PL electrode with multiple cyclic voltammetry measurements first in pH 10.9 then in pH 7.4, we found a dramatic decrease of anodic peak current in comparison with those continuously measured in pH 7.4, suggesting the possible damage of PEDOT film under pH 10.9. It has been pointed out that the electrochemical performance of PEDOT:PSS modified electrode under the pH above 10 would cause the over-oxidation of PEDOT film that might result from the breakage of the dioxane ring, therefore reducing the conductivity of PEDOT film [43]. That could be the major cause for the drop of anodic current while pH was changed from 7.4 to 10.9 in Figure 5. In addition, the temperature has insignificant effects on the anodic peak current, which slightly increased with temperature in the range of 17 to 37 °C, then decreased at 45 °C (Appendix A).

### 3.5. The Effect of Synthesis Cycle for PEDOT Film on the Performance

A study on the measurement of HQ with a PEDOT modified platinum electrode has demonstrated that the anodic peak current was significantly enhanced by increasing the PEDOT film thickness due to the increase of effective surface area [37]. Similar results were reported for the poly(thionine) modified glassy carbon electrode [14]. In this study, we found that the increase of the cyclic voltammetry cycle for PEDOT synthesis resulted in the increase of the oxidative detection current, however, both anodic and cathodic peaks gradually lost their peak symmetry and exhibited a tailed voltammogram where the waves did not quickly return back to their baseline, as shown in Figure 6A. This phenomenon might suggest that increase in PEDOT film thickness might result in the slower mass transport of HQ for the anodic part and BQ for the cathodic part to the surface of the electrode. It is necessary to be declared whether it is caused by increasing the size of the diffusion layer as well as film resistance due to more cycles applied for PEDOT synthesis. Considering the limited contribution of the anodic peak current from the increasing of cyclic voltammetry synthesis cycles (Figure 6B), we recommended six cycles for PEDOT synthesis in this study.

In addition, the size of the diffusion layer above the electrode surface could be determined by the employed potential scan rate. In general, the diffusion layer will be continuously built up and become thicker above the electrode with a slow potential scan rate, which may result in a lesser concentration gradient between the bulk solution and the surface of the electrode and thus a smaller mass flux of the target compound, such as HQ in this study. Consequently, the magnitude of the electrochemical current that was proportional to the mass transport rate of HQ towards the electrode was developed to be larger and larger with the increase in the potential scan rate, as shown in Figure 7A. However, it was also accompanied by a larger peak-to-peak separation with a fast scan rate, implicating that the rate of electrochemical kinetics on the PEDOT/PL electrode was still not very efficient to achieve the equilibrium at the solution-electrode interface. Figure 7B demonstrates the linear correlation between the anodic peak current and the square root of scan rate, suggesting a diffusion-controlled process. This would lead us to future studies on the improvement of electrochemical kinetics of HQ on the PEDOT/PL electrode through either the enhancement of the electrochemical activity of PEDOT film or the acceleration of the chemical reaction on the surface of PEDOT film.

### 3.6. Stability of PEDOT/PL for Multiple Measurements

During our routine experiments, the PEDOT/PL electrode was found to be reusable for multiple measurements. As shown in Figure 8A, the PEDOT/PL electrode was quite stable in detecting 6 mM of HQ for 10 individual tests, and the electrode was thoroughly washed with distilled water three times after each test and stored under room temperature for the next test. The reduction of anodic peak current could be controlled within 10% for the first 10 tests and 20% for the next 20 tests. With the modification of PEDOT film and the optimization of the wash process, the stability of PEDOT/PL electrode performance could be further improved.

### 3.7. Potential Interference on the Measurement

Since HQ is easily oxidized to form catechol, they are sometimes coexisting in the environment or human body, it is, therefore, necessary to understand the interference of catechol on the HQ measurement. As shown in Figure 9A, there was no obvious anodic peak for catechol at the potential near 0.1 V, which was assigned for the anodic peak of HQ. In addition, mixing 6 mM HQ and 0.05 mM catechol, we could only identify the oxidation of HQ at the potential of 0.1 V without notable reduction. Meanwhile, other naturally electroactive species in blood, such as glucose, vitamin C, uric acid, and others, can be easily oxidized on the electrode, therefore generating potential interferences in the accurate detection of the target compound. In this study, we investigated the influence of glucose, vitamin C, and uric acid on the detection of 6 mM of HQ. The results were shown in Figure 9, where 0.5 mM of vitamin C and 10 mM of glucose resulted in a 1.31 and 3.50% reduction of anodic peak current, respectively. However, 0.02 mM of uric acid generated about 17.33% reduction of anodic peak current in comparison with that of HQ alone. To effectively eliminate the potential interferences from uric acid and others, researchers have recommended employing Nafion and polymer micromembranes to build up permselective barriers [44,45,46]. Nevertheless, our results have indicated the potential applications of PEDOT/PL electrodes for environmental and clinical measurement of HQ.

### 3.8. The Linear Sweep Voltammetry (LSV) Measurement of HQ with the PEDOT/PL Electrode

Considering the inevitable influences from the diffusion resistance and possible adsorption of hydroquinone and benzoquinone on cyclic voltammetry measurements. We then employed the simple and quick linear sweep voltammetry (LSV). From Figure 10A, we could clearly notice the dramatic increase of the oxidative peak current by increasing the concentration of hydroquinone. Figure 10B shows that the anodic peak current had a nice linear correlation with the concentration of hydroquinone. The sensitivity was about 901.7 μA·mM^−1^·cm^−2^, which was increased by 22.8% in comparison with CV measurement.

### 3.9. The Practical Applications and Future Research Perspectives of Conductive PEDOT in Biosensors

Conductive polymers have gained tremendous attention and inspired extensive studies in scientific communities, especially since the Nobel Prize in Chemistry 2000 was awarded to Alan J. Heeger, Alan G. MacDiarmid, and Hideki Shirakawa who were pioneers in the discovery and development of conductive polymers. Among them, poly(3,4-ethylenedioxythiophene) or PEDOT is the most amazing conductive polymer due to its intrinsic features of low bandgap, great conductivity, pronounced electrochromic activity, long-term air stability, and many promising applications in various fields and it is easily synthesized [47,48]. Previously, we have successfully employed horseradish peroxidase (HRP) to synthesize PEDOT with good electrochemical properties, excellent solubility, and biocompatibility [49,50]. Recently, we have also developed a green chemical process by using non-thermal plasma activated hydrogen peroxide to promote the synthesis of PEDOT and demonstrated the degradability of PEDOT [51,52]. Meanwhile, we have employed PEDOT to modify a platinum electrode to construct biosensors for sensing hydrogen peroxide and glucose [31,32,35]. So far, PEDOT has been proposed for a great variety of promising applications in bioelectronics, conductive hydrogel, and other functional materials [53,54,55,56].

The original goal of this study was to modify the graphene or its derivatives for the construction of a hydroquinone biosensor. In modern materials, graphene is sometimes considered a spectacular material, possessing remarkable mechanical properties, impressive thermal conductivity, and quick electron mobility [57], but it is normally quite expensive and difficult to use for mass production. On the other hand, graphene oxide (GO) can be easily made from abundant graphite and tuned with oxygen-containing groups, but its conductivity is somehow not satisfactory. Fortunately, the intrinsic properties of graphene could be regained by transferring GO to reduced graphene oxide (rGO). Due to the lack of functional groups that are ready for the automatic covalent binding of functional enzymes, it has to go through the encapsulation of enzymes in a porous matrix of the chemical crosslinking of enzymes onto the rGO coated electrode surface [58,59]. To fabricate a successful biosensor, golden nanoparticles could electrochemically deposit onto the rGO membrane with a time-consuming procedure [45]. Reproducibility is always the main challenge in developing rGO-based electrochemical sensors [58,60,61]. On the other hand, there are abundant sulfur atoms in a PEDOT molecular structure that are ready for the stable immobilization of golden nanoparticles, which can then immobilize a variety of biomolecules with thiol groups (-SH) to construct various biosensors [31,32,33]. Modification of a platinum electrode with the conductive PEDOT film has also been reported to effectively eliminate the evitable hydrogen ion reduction or water oxidation at slightly negative or positive potentials [33], and to improve the sensitivity of a biosensor [32,34,35,36]. Therefore, the modification of GO or rGO with the conductive PEDOT will open up an era of fabricating more effective sensors for various applications.

Recently, pencil lead (PL), which is usually made from stacked graphite, has drawn much attention in developing low-cost sensors because of its ready availability and ease of modification [62,63]. In this study, we modified an inexpensive PL with electrochemically synthesized PEDOT film and explored its application in the detection of hydroquinone (HQ). In comparison with a bare PL electrode, our results have demonstrated that the PL electrode modified with electrochemically synthesized PEDOT film had the advantages of reducing redox potentials, accelerating electron transfer, and minimizing the adsorption of HQ and its reduced product on the electrode. Besides, PEDOT film provides a platform for the stable capture of golden nanoparticles by a self-assembly procedure, and then the biomolecules with S-H functional groups. So far, the main challenge of the PEDOT/PL electrode is to eliminate the interferences from the electroactive species in human blood. Fortunately, the interferences can be effectively eliminated by a permselective membrane [44,45,46]. In conclusion, PEDOT/PL sensors have promising perspectives in fabricating low cost and effective sensors.

## 4. Conclusions

HQ has been wildly used in industry and many livelihood products, scientific researches have implied that HQ, either in nature or derived from body metabolism, has a potential impact on human health. Our goal is to construct a very low-cost but quite effective electrode for monitoring hydroquinone (HQ). In comparison with the use of a platinum electrode as a substrate, the PEDOT film on a PL electrode was very stable and ready for multiple measurements. Both cyclic voltammetry (CV) and linear sweep voltammetry (LSV) demonstrated that the PEDOT/PL electrode possessed competitive sensitivity, high selectivity, and good stability in sensing HQ with fine correlations between the anodic peak current of HQ oxidation and the concentration of HQ. Some electroactive compounds, such as glucose and vitamin C, naturally found in human blood have been examined to show negligible interference on the HQ detection. Although uric acid has a significant interference in HQ detection, it could be eliminated by a permselective membrane as recommended by the literature [44,45,46]. Nevertheless, the performance of the PEDOT/PL electrode could be further improved by modifying or functioning PEDOT film and optimizing detection condition or procedure. Therefore, the extremely cheap, easily available, and disposable pencil lead (PL or graphite) is a suitable electrode for the construction of various chemical sensors or biosensors.

## Figures and Tables

**Figure 1 polymers-13-00343-f001:**
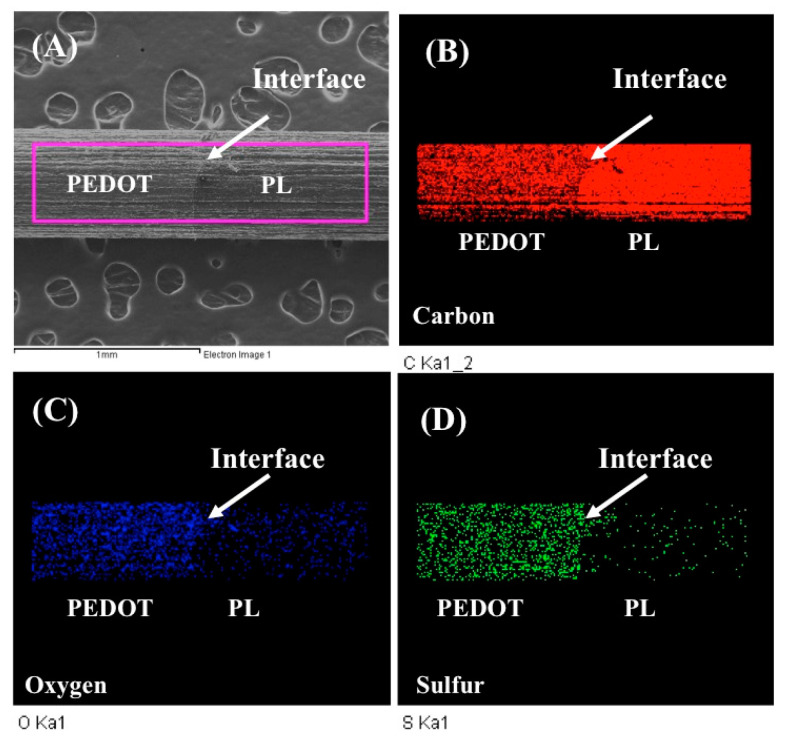
The surface images of PL and PEDOT/PL electrodes by SEM. (**A**) the interface between PEDOT film and uncoated PL. (**B**–**D**) the element mapping of carbon, oxygen, and sulfur for PL and PEDOT film, respectively.

**Figure 2 polymers-13-00343-f002:**
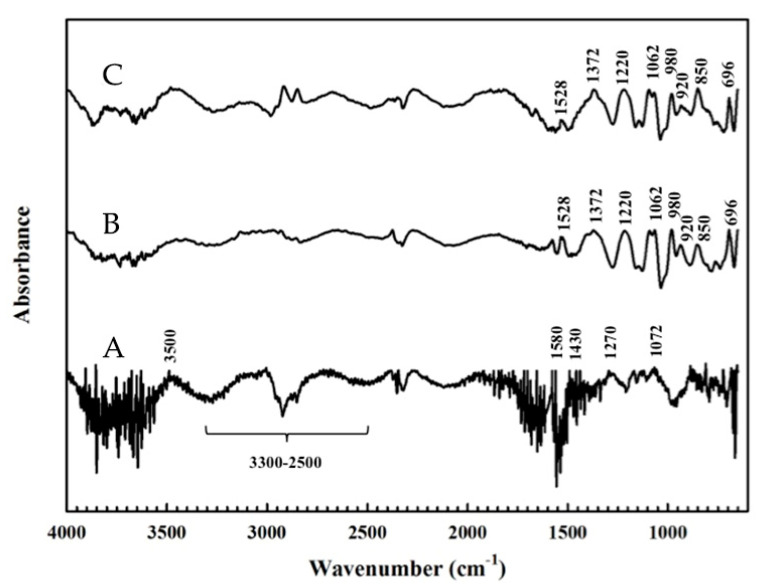
FTIR spectra of (**A**) PL electrode, (**B**) PEDOT film on a PL electrode, and (**C**) PEDOT/PL electrode after HQ measurement.

**Figure 3 polymers-13-00343-f003:**
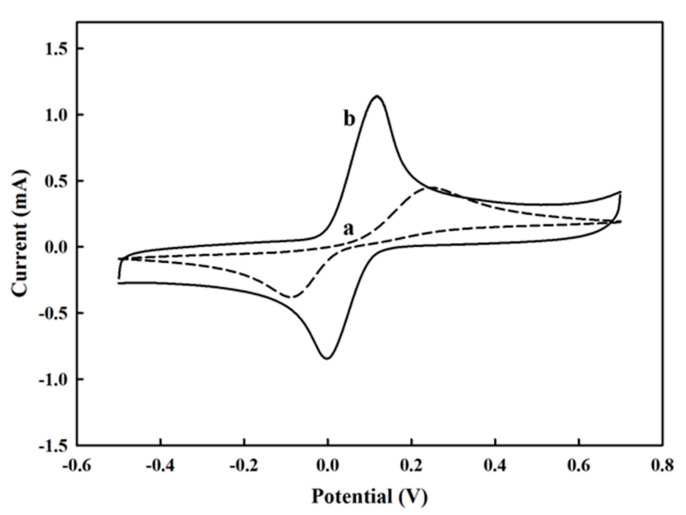
The cyclic voltammetry profiles of (**a**) a bare PL electrode and (**b**) a PEDOT/PL electrode in response to 6 mM of hydroquinone in PBS buffer. The potential was swept from −0.5–0.7 V with the rate of 100 mV s^−1^.

**Figure 4 polymers-13-00343-f004:**
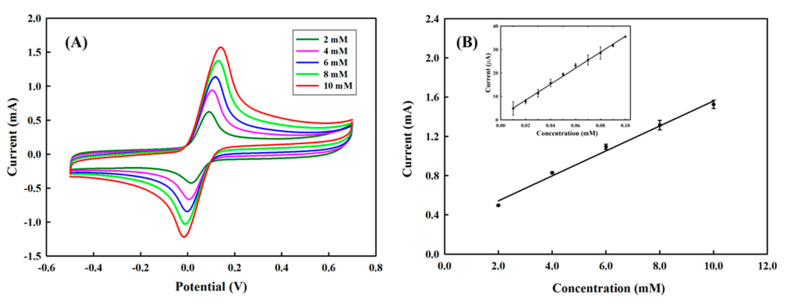
(**A**) The cyclic voltammetry profiles of PEDOT/PL electrode in response to different concentrations of hydroquinone (2 to 10 mM). (**B**) The linear correlation between the anodic peak current and the concentration of hydroquinone. The inlet indicated the linear correlation for a low concentration of hydroquinone (10–100 μM). The measurements were carried in 100 mM of PBS buffer. The potential was swept from −0.5–0.7 V with the rate of 100 mV s^−1^.

**Figure 5 polymers-13-00343-f005:**
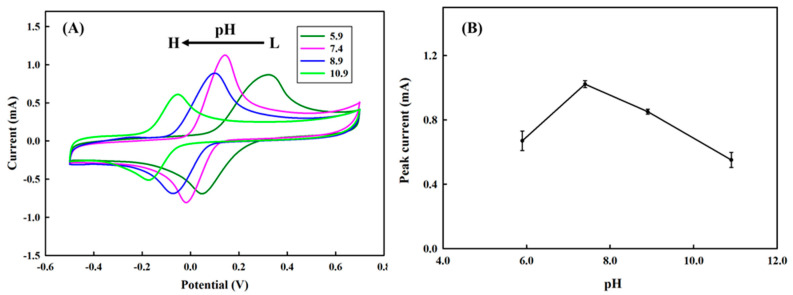
(**A**) The effect of pH on the cyclic voltammetry profiles for the detection of 6 mM of hydroquinone in 100 mM of PBS buffer, where pH 5.9 (green), pH 7.4 (pink), pH 8.9 (navy), and pH 10.9 (limo). The potential was swept from −0.5–0.7 V with the rate of 100 mV s^−1^. (**B**) The effect of pH on the anodic peak current.

**Figure 6 polymers-13-00343-f006:**
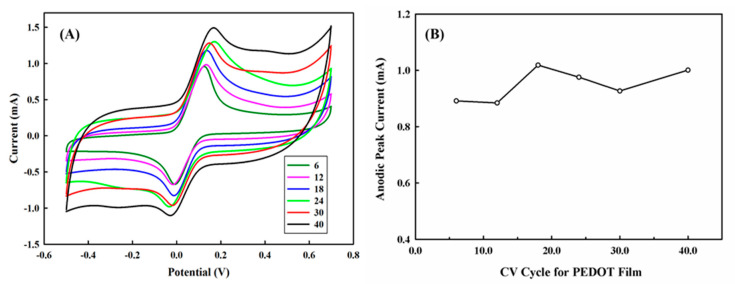
(**A**) The effect of cyclic voltammetry synthesis cycles, 6–40 cycles, on the performance of PEDOT/PL electrode for the detection of 6 mM of hydroquinone in 100 mM of PBS buffer. The potential was swept from −0.5–0.7 V with the rate of 100 mV s^−1^. (**B**) The effect of CV cycles on the anodic peak current.

**Figure 7 polymers-13-00343-f007:**
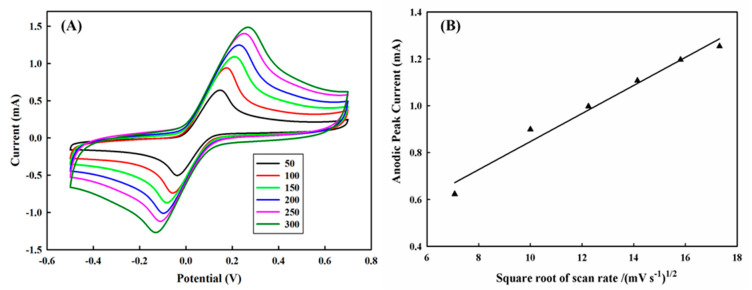
(**A**) The effect of cyclic voltammetry scan rate on the performance of PEDOT/PL electrode for the detection of 6 mM of hydroquinone in 100 mM of PBS buffer. The potential was swept from −0.5–0.7 V with different scan rate (**B**) The effect of the square root of scan rate on the anodic peak current.

**Figure 8 polymers-13-00343-f008:**
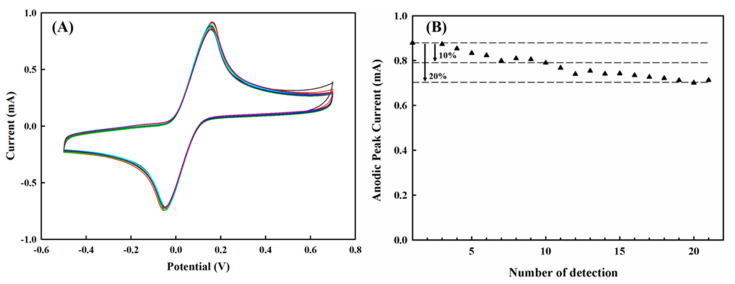
The stability analysis of PEDOT/PL electrode for the detection of 6 mM of hydroquinone in 100 mM of PBS buffer. The potential was swept from −0.5–0.7 V with the rate of 100 mV s^−1^. (**A**) The cyclic voltammetry profiles for ten individual tests. The same electrode was thoroughly washed with distilled water three times after each test. (**B**) The anodic peak current for each measurement.

**Figure 9 polymers-13-00343-f009:**
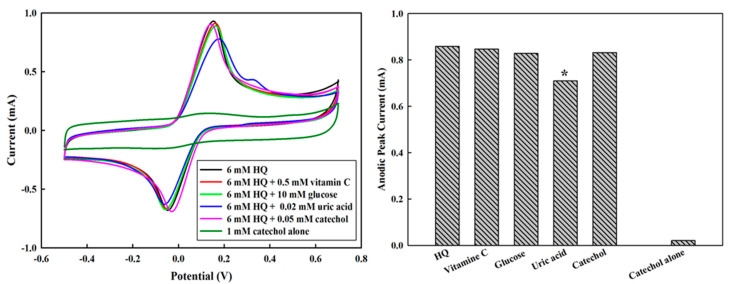
Potential interference on the measurement. (**A**) the cyclic voltammetry profiles for 6 the detection of 6 mM HQ in the presence of vitamin C, glucose, uric acid, and catechol. The potential was swept from −0.5–0.7 V with the rate of 100 mV s^−1^. (**B**) The influence of interference on the anodic peak currents of CV for 6 mM HQ, where * denotes the significant reduction in comparison with that for HQ only (first bar).

**Figure 10 polymers-13-00343-f010:**
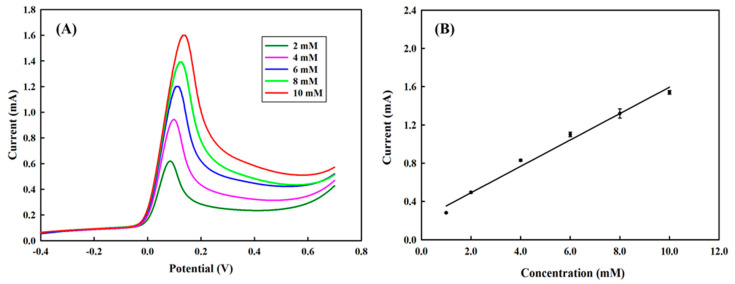
(**A**) The LSV profiles of PEDOT/PL electrode in response to different concentrations of hydroquinone (2 to 10 mM). The measurements were carried in 100 mM of PBS buffer. The potential was swept from −0.5–0.7 V with the rate of 100 mV s^−1^. (**B**) The linear correlation between the anodic peak current and the concentration of hydroquinone.

## Data Availability

The data presented in this study are available on request from the corresponding author.

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
