# Peer review of "Fabrication of an Extremely Cheap Poly(3,4-ethylenedioxythiophene) Modified Pencil Lead Electrode for Effective Hydroquinone Sensing"

_polymers, 2021, doi:10.3390/polym13030343_

Round 1

Reviewer 1 Report

The reviewed manuscript presents the results of extremely cheap poly(3,4-ethylenedioxythio- phene) modified pencil lead electrode for effective hydroqui- none sensing This research is novel and author write it well. However, the purposefulness and effectiveness of using this system raise serious doubts.The  manuscript should be improved before being considered for publication. I will accept this manuscript for publication in polymer after minor changes.

  1. Please draw a graphical abstract so it will be easy for reader to get idea directly from GA.
  2. Abstract: This part of the text is really hard to understand, it does not present the most important conclusions obtained from the research in a concentrated manner, but at the same time understandable for a potential reader. It does not at all encourage a study of the manuscript. The authors must thoroughly edit this text.
  3. Introduction: The authors cite the relevant works, but for the most important ones, there is lack of deeper, more detailed description that would allow for a later comparison of the system developed by the authors with the similar and previously used ones.
  4. Discussion: This part requires a thorough development. The authors should clarify the   signalized doubts. They should to demonstrate the advantages and disadvantages of theproposed system against the background of similar systems described earlier. The authors should also present their suggestions related to possible possibilities of practical application of the described solution.
  5. In general, English language and style must be improve in the whole manuscript, mainly in sections such as the introduction and the experimental section.

Reviewer 2 Report

The manuscript entitled Fabrication of an extremely cheap poly(3,4-ethylenedioxythiophene) modified pencil lead electrode for effective hydroquinone sensing submitted to Polymers Journal.

The concept of the manuscript is novel, fits and suitable to publish in Polymers Journal. This manuscript is generally well written and clearly presented however still need to address many comments and thus require substantial major revision before its acceptance.

  • Provide a nice graphical abstract representing the overview of the MS with key highlights.
  • In abstract authors should mention should mention the values of results and importance of research work in one or two sentences. Also mention somewhere the full form of abbreviations used in the manuscript.
  • In the introduction section, write the novelty of the work and the problem statement clearly. The defined research objectives should be mentioned ta the end of introduction. Substantial discussion about the economics of the developed process is essential.
  • Section 2.2 Give details of operational conditions and detailed procedures. Where is Fig1?
  • Give values to the peak of FTIR. Substantial discussion of peaks and their comparison with the literature is expected during revision.
  • Have authors studied Effect of temperature, give details. In potential of interference studies why authors selected these chemical need to discuss in the section.
  • Studies of real sample if provided then this study become more impressive.
  • Importance and advantages of this studies in medical field need to describe somewhere.
  • Write the practical applications and future research perspectives and challenges by adding a new section before conclusions
  • The conclusion of the study is not discussed with the specific output obtained from the study, it could be modified with precise outcomes with a take home message.
  • English and grammar mistakes are present. The author should check the manuscript by native English Speaker to improve the quality of the manuscript.

Round 2

Reviewer 2 Report

Authors have revised the manuscript according to reviewers comments. However, still some points need revision before its acceptance

1) In graphical abstract electron abbreviation should be in standard format. In my opinion the arrow position should be in reverse direction to understand the research work easily.

2)FTIR peak values need to be denoted.

3) In the potential of interference studies why authors selected these chemicals need to be discussed in the section. The details should be mentioned in the text.

4) Write the practical applications and future research perspectives and challenges not in the conclusion section but by adding a new section.

5) Add a few recent references of the year 2018-2020 during revision.

Also provide the detailed procedure and operational conditions  of Characterizations of PEDOT/PL
